# Para-Substituted *O*-Benzyl Sulfohydroxamic Acid Derivatives as Redox-Triggered Nitroxyl (HNO) Sources

**DOI:** 10.3390/molecules27165305

**Published:** 2022-08-19

**Authors:** Yueming Long, Zijun Xia, Allison M. Rice, S. Bruce King

**Affiliations:** Department of Chemistry, Wake Forest University, Winston-Salem, NC 27101, USA

**Keywords:** nitroxyl (HNO), Piloty’s acid, redox triggered 1, 6 elimination, gasotransmitters, redox signaling, aromatic nitro/azide reduction, aromatic boronate oxidation

## Abstract

Nitroxyl shows a unique biological profile compared to the gasotransmitters nitric oxide and hydrogen sulfide. Nitroxyl reacts with thiols as an electrophile, and this redox chemistry mediates much of its biological chemistry. This reactivity necessitates the use of donors to study nitroxyl’s chemistry and biology. The preparation and evaluation of a small library of new redox-triggered nitroxyl sources is described. The condensation of sulfonyl chlorides and properly substituted *O*-benzyl hydroxylamines produced *O*-benzyl-substituted sulfohydroxamic acid derivatives with a 27–79% yield and with good purity. These compounds were designed to produce nitroxyl through a 1, 6 elimination upon oxidation or reduction via a Piloty’s acid derivative. Gas chromatographic headspace analysis of nitrous oxide, the dimerization and dehydration product of nitroxyl, provides evidence for nitroxyl formation. The reduction of derivatives containing nitro and azide groups generated nitrous oxide with a 25–92% yield, providing evidence of nitroxyl formation. The oxidation of a boronate-containing derivative produced nitrous oxide with a 23% yield. These results support the proposed mechanism of nitroxyl formation upon reduction/oxidation via a 1, 6 elimination and Piloty’s acid. These compounds hold promise as tools for understanding nitroxyl’s role in redox biology.

## 1. Introduction

The one-electron reduction and protonation of nitric oxide (NO), a well-known biological signaling agent characterized as a gasotransmitter, formally produces nitroxyl (HNO) [1]. These structural/electronic differences give HNO a distinct chemistry from NO, as HNO dimerizes to hyponitrous acid (H_2_N_2_O_2_) that dehydrates to nitrous oxide (N_2_O) [2]. This reactivity necessitates the use of HNO donors and highlights the extreme electrophilic and oxidizing character of HNO [3]. HNO demonstrates different biological properties from NO [4,5], and at least three drugs that chemically release HNO have been used clinically or evaluated in trials for the treatment of cancer (hydroxyurea), alcoholism (cyanamide) and congestive heart failure (Cimlanod), showing the clinical potential of HNO donors [6,7,8]. Much of our understanding of the pharmacology and therapeutic potential of HNO comes from experiments utilizing HNO donors [3,9]. The most common HNO donors include Angeli’s salt (AS, Na_2_N_2_O_3_) and Piloty’s acid (PA, PhSO_2_NHOH), which are commercially available solids that rapidly and cleanly release HNO under neutral conditions [3,9].

In addition to dimerization, HNO reacts with thiols, generating a N-hydroxysulfenamide intermediate that can rearrange to a sulfinamide or further react with more thiol to yield a disulfide and hydroxylamine [10]. HNO similarly reacts with hydrogen sulfide (H_2_S), a second recognized gasotransmitter [1], to give short-chain hydrogen polysulfides (H_2_S*_n_*) or S_8_ depending on the relative concentrations of HNO and H_2_S [11]. Based on this chemistry, HNO can influence thiol-mediated biochemistry and potentially H_2_S-controlled reactions. For example, HNO inhibits enzymes with active site cysteines, such as aldehyde dehydrogenase (AlDH) and glyceraldehyde 3-phosphate dehydrogenase (GAPDH) [12,13]. HNO modifies cardiac myofilament proteins affecting myocardial contractility by increasing calcium cycling and sensitizing myocardial tissue responsiveness to calcium [14,15]. HNO directly activates the chemosensory TRPA1 channel [16]. In bacteria, the addition of HNO to *Staphylococcus aureus* increases the formation of the persulfide (RSSH) of the predominant low-molecular-weight thiol, bacillithiol (BSH) [17]. This oxidized thiol derivative may control transcription factors responsible for directing the overall sulfur metabolism in these bacteria [18]. Under oxidative conditions, the addition of HNO to *Bacillus subtilis* synergistically enhances hydrogen peroxide (H_2_O_2_)-mediated cell killing. Angeli’s salt (AS), which decomposes to HNO and nitrite (NO_2_^−^) at neutral pH, was used as the HNO source in these redox investigations in *B. subtilis* [3,19].

Redox-triggered HNO donors may possess value in probing biological crosstalk between HNO and other small redox active signaling agents, such as H_2_S or H_2_O_2_. The use of AS as a HNO donor is limited by the co-production of NO_2_^−^, a relatively fast and pH-insensitive release rate (t_1/2_~2.8 min, pH 4–8) and limitations in the synthesis of new donors [3]. Piloty’s acid (PA), another common HNO donor, decomposes to HNO and phenyl sulfinic acid (PhSO_2_H) at a neutral pH [3]. The aryl portion of PA tolerates structural modification while still supporting HNO release, which allows for the installation of designed redox-sensitive elements to initiate a 1, 6-elimination [20]. Figure 1 shows Piloty’s acid derivatives designed to release HNO via a 1, 6-elimination mechanism upon exposure to either reductants or oxidants. These processes convert the azide, nitro or boronate ester groups into either the aniline or phenol derivative that should decompose to Piloty’s acid with the release of *p*-quinone methide or its imine (Figure 1). 

Similar constructs have found extensive use for the detection or generation of other signaling agents, such as H_2_S, but, to the best of our knowledge, have not been applied to HNO release under redox conditions [21]. A hydrogen-peroxide-based prodrug system that releases a structurally similar hydroxamic acid as a histone deacetylase inhibitor has been described [22]. Such compounds permit HNO formation under specific redox conditions and would form the basis of an improved understanding of the role that HNO plays in redox biochemistry. We describe the preparation and characterization of a small library of redox-triggered HNO donors and an evaluation of their ability to produce HNO under specific redox conditions.

## 2. Results

### 2.1. Synthesis

Possible redox-triggered HNO donors (**1a**–**d** and **2a**–**c**) were prepared with a 27–79% yield by the condensation of the properly substituted O-benzyl hydroxylamine derivative with either *p*-toluene or methyl sulfonyl chloride under basic conditions (Figure 2) [23,24]. The variable yields likely arise from mixtures of O and N-mono alkylated products and N, O-dialkylated products as reported [23,24].

This sequence followed by extraction generally gave the desired products with excellent purity, as judged by proton and carbon nuclear magnetic resonance (NMR) spectroscopy and mass spectrometry (MS). Individual compounds could be further purified by recrystallization or silica gel flash chromatography if necessary.

The substituted *O*-benzyl hydroxylamines required in Figure 1 were purchased (R = -H or -NO_2_) or prepared using reported literature procedures for the azide and the pinacol-derived boronate ester [22,25]. Figure 3 summarizes the preparation of the azide-containing hydroxylamine (**6**) from *p*-toluidine through a four-step sequence that features diazotization/azide substitution, bromination, N-hydroxyphthalimde substitution to install the N-O bond and hydrazine deprotection through intermediates **3**–**5** (Figure 3) [25].

The Mitsonobu coupling of N-hydroxyphthalimide with the commercially available pinacol ester of 4-hydroxymethylphenylboronic acid followed by hydrazine deprotection yields the boronate-ester-substituted hydroxylamine (**8**) via **7** (Figure 3) [22]. These intermediates were characterized by NMR spectroscopy and MS and purified by normal phase flash chromatography (see Appendix A).

### 2.2. HNO Donation Ability

Piloty’s acid (PhSO_2_NHOH), a common HNO donor, decomposes to HNO and phenyl sulfinic acid (PhSO_2_H) at a neutral pH. Compounds **1b**–**d** and **2b**,**c** were designed to decompose upon either reduction or oxidation as depicted in Figure 1 to yield a reactive intermediate that should release a Piloty’s acid derivative (*p*CH_3_C_4_H_6_SO_2_NHOH or CH_3_SO_2_NHOH) that fragments to HNO and the corresponding sulfinic acid. The gas chromatographic (GC) headspace measurement of nitrous oxide (N_2_O), the dimerization and dehydration product of HNO, provides a rapid and simple measure of HNO formation from these transformations [26]. The decomposition of the PA derivative *p*CH_3_C_4_H_6_SO_2_NHOH, a known HNO donor, in methanol/buffer generates 76% of N_2_O by this measure after 24 h (Table 1). The sensitivity of N_2_O formation to the addition of glutathione (GSH), which rapidly reacts with HNO blocking N_2_O production [10], provides evidence for HNO’s intermediacy (Table 1). Compounds **1a** and **2a** do not contain reduction or oxidation-sensitive functional groups, preventing the proposed decomposition to a Piloty’s acid derivative, and do not produce N_2_O under these conditions (Table 1). 

The incubation of **1b-d** and **2b-c** in a methanol/100 mM PBS buffer showed ~1% N_2_O formation, indicating essentially no formation of HNO over 24 h from these compounds, which was as expected, in the absence of any reducing/oxidizing agents. The formation of small amounts of N_2_O could arise from the slow hydrolysis/methanolysis of the benzyl group producing *p*CH_3_C_4_H_6_SO_2_NHOH or CH_3_SO_2_NHOH or from the presence of small amounts of these sulfohydroxamic acids in these samples, which could form from the condensation of residual hydroxylamine potentially present in the commercial or synthetic O-benzylhydroxylamine derivatives. 

Table 2 summarizes N_2_O formation at 24 h from the treatment of **1a**–**d** and **2a**–**c** with various reducing and oxidizing agents. The addition of sodium borohydride, a reducing agent capable of nitro-to-amine reduction [27], to compounds **1b** and **2b** generates N_2_O with a 25 and 51% yield, respectively, providing evidence for initial HNO formation (Table 2). The lower observed amounts of N_2_O from **1b** may reflect the poor solubility of **1b**, which did not completely dissolve in 1:1 CH_3_OH:H_2_O (2 mL), possibly due to the presence of two para-substituted aromatic rings, including one with the polar nitro group. Compound **1b** dissolved with the addition of another 0.5 mL of CH_3_OH (1.5:1 CH_3_OH:H_2_O, 2.5 mL), and the results reported in Table 2 for **1b** were obtained under these conditions. Thin-layer chromatography (TLC) and an MS analysis of the reduction of **1b** with sodium borohydride provides evidence for the formation of *p*-amino benzyl alcohol, the product of water addition to the imine of *p*-quinone methide. The addition of sodium borohydride/copper (II) sulfate, a mixture known to reduce azides to amines [28], to **1c** and **2c** produces 89 and 92% N_2_O, respectively (Table 2). These results support Figure 1 and suggest a reduction of the nitro and azide groups to the amine followed by decomposition to the Piloty’s acid derivative. As expected, the treatment of **1a** and **2a** with sodium borohydride or sodium borohydride/copper (II) sulfate under these conditions did not generate N_2_O (Table 2). 

The incubation of **1c** and **2c** with GSH (2.5 or 5 equivalents) does not produce N_2_O, but the treatment of **1c** and **2c** with H_2_S (2.5 equivalents) generated small reproducible amounts (6% and 5%, respectively) of N_2_O. These results suggest H_2_S-mediated azide reduction to the amine as described, followed by HNO formation [29]. The lower amounts of N_2_O observed are likely due to the competition between HNO dimerization and H_2_S addition. Increasing the amounts of H_2_S to five equivalents abolished N_2_O formation, supporting this explanation. Table 2 also shows hydrogen peroxide (H_2_O_2_)-mediated N_2_O release from **1d**, a boronate-containing compound designed to release HNO via boronate oxidation to the phenol followed by decomposition. The incubation of **1d** with H_2_O_2_ resulted in the formation of N_2_O with a 23% yield, presumably indicating the formation of HNO upon oxidation to the phenol. The treatment of **1a** and **2a** with H_2_O_2_ did not produce N_2_O (Table 2). Table 2 also shows that the amount of N_2_O produced increases from the 1 h to 24 h measurements, likely reflecting the kinetics of these model oxidations and reductions.

Bacterial nitroreductases using NADH as a co-substrate act as competent reducing agents of aromatic nitro groups, with the subsequent release of desired compounds via a 1, 6-elimination [30,31]. The treatment of **1b** or **2b** with *Escherichia coli* nitroreductase (Sigma) in the presence of NADH failed to generate N_2_O as initially expected. A previous report indicates that NADH reduces HNO to hydroxylamine, thus blocking N_2_O formation and suggesting that the lack of observed N_2_O in these experiments results from HNO reduction by the NADH co-substrate [32]. Further investigation will be necessary to define the practicality of nitroreductase-triggered HNO donors, but these results suggest that HNO may exert a portion of effects through NADH/NADPH depletion. 

## 3. Materials and Methods

All materials and solvents used for extraction and purification were purchased from commercial vendors and used as received. ^1^H and ^13^C NMR spectra were recorded using a Bruker Avance 400 MHz NMR spectrometer. Mass spectra were obtained using a Bruker Amazon SL ion trap. An Agilent Technologies 7890A gas chromatograph equipped with a micro-electron capture detector and a 30 m × 0.32 m (25 μm) HP-MOLSIV capillary column was used for the gas chromatographic analysis of N_2_O. CAUTION: Any experiments preparing alkyl or aryl azides should be performed in a well-ventilated fume hood and behind a blast shield. Sodium azide should not be mixed with strong acids. 

### 3.1. Synthesis of Piloty’s Acid Derivatives

The Piloty’s acid derivatives were prepared by sulfonyl chloride condensation with an O-benzylhydroxylamine derivative using a modified literature procedure [33].

***N*-(benzyloxy)-4-methylbenzenesulfonamide (1a).***O*-benzylhydroxylamine hydrochloride (287 mg, 1.80 mmol) was added to a solution of *p*-toluenesulfonyl chloride (378 mg, 1.98 mmol) in pyridine (5 mL). The reaction mixture reacted overnight at room temperature and was quenched with 2M HCl (25 mL) and extracted with ethyl acetate (25 mL). The organic layer was washed with water (10 mL), brine (10 mL), dried over MgSO_4_ and concentrated using rotary evaporator to give 396 mg, 79.4% yield of **1a** as white solid. ^1^H NMR (400 MHz, CDCl_3_): δ 7.82 (d, *J* = 8.0 Hz, 2H), 7.33–7.35 (m, 6H), 6.88 (s, 1H), 4.97 (s, 2H), 2.44 (s, 3H). ^13^C NMR (101 MHz, CDCl_3_): δ 148.02, 145.33, 142.47, 133.33, 129.90, 129.59, 128.56, 123.75, 77.79, 21.72. ESI-MS (positive mode) (*m*/*z*) calculated mass for C_14_H_14_NO_3_SNa [M+Na]^+^ 300.34, found mass 300.13. 

**4-methyl-*N*-((4-nitrobenzyl)oxy)benzenesulfonamide (1b)**. 1-[(Aminooxy)methyl]-4-nitrobenzene hydrochloride (368 mg, 1.80 mmol) was added to a solution of *p*-toluenesulfonyl chloride (378 mg, 1.98 mmol) in pyridine (5 mL). The reaction mixture reacted overnight at room temperature and was quenched with 2M HCl (25 mL) and extracted with ethyl acetate (25 mL). The organic layer was washed with water (10 mL), brine (10 mL), dried over MgSO_4_ and concentrated under reduced pressure to give 428 mg, 73.8% yield of **1b** as white solid. ^1^H NMR (400 MHz, CDCl_3_): δ 8.20 (d, *J* = 8 Hz, 2H), 7.80 (d, *J* = 8 Hz, 2H), 7.49 (d, *J* = 8 Hz, 2H), 7.35 (d, *J* = 8 Hz, 2H), 6.96 (s, 1H), 5.09 (s, 2H), 2.45 (s, 3H). ^13^C NMR (101 MHz, CDCl_3_): δ 147.96, 145.32, 142.56, 133.31, 129.89, 129.54, 128.55, 123.72, 79.74, 21.71. ESI-MS (positive mode) (*m*/*z*) calculated mass for C_14_H_14_N_2_O_5_SNa [M+Na]^+^ 345.06, found mass 345.20.

***N*-((4-azidobenzyl)oxy)-4-methylbenzenesulfonamide (1c)**. *p*-Toluenesulfonyl chloride (2.24 g, 11.8 mmol) was added to a solution of *O*-(4-azidobenzyl) hydroxylamine (1.78 g, 10.8 mmol) in pyridine (30 mL). The reaction mixture was allowed to stir overnight at room temperature and quenched with 2M HCl (150 mL) and extracted using ethyl acetate (150 mL). The organic layer was washed with water (100 mL), brine (100 mL), dried over MgSO_4_ and concentrated under reduced pressure to give 1.32 g, 38.4% yield of **1c** as a light yellow solid. ^1^H NMR (400 MHz, CDCl_3_): δ 7.73 (d, *J* = 8 Hz, 2H), 7.26 (d, *J* = 8 Hz, 4H), 6.93 (d, *J* = 8 Hz, 2H), 6.85 (s, 1H), 4.87 (s, 2H), 2.37 (s, 3H). ^13^C NMR (101 MHz, CDCl_3_): δ 145.00, 140.53, 133.60, 131.96, 131.04, 129.79, 128.55, 119.09, 78.66, 21.69. ESI-MS (positive mode) (*m*/*z*) calculated mass for C_14_H_14_N_4_O_3_SNa [M+Na]^+^ 341.34, found mass 341.28.

**4-methyl-*N*-((4-(4,4,5,5-tetramethyl-1,3,2-dioxaborolan-2-yl)benzyl)oxy)benzenesulfonamide (1d)**. *p*-Toluenesulfonyl chloride (340 mg, 1.79 mmol) was added to a solution of *O*-(4-(4,4,5,5-tetramethyl-1,3,2-dioxaborolan-2-yl)benzyl)hydroxylamine (410 g, 1.64 mmol) in pyridine (5 mL). The reaction mixture reacted overnight and concentrated under reduced pressure with the addition of excess toluene. The resulting product was acidified using 2M HCl and extracted with chloroform (25 mL). The organic layer was washed with water (30 mL), brine (30 mL), dried over MgSO_4_ and concentrated under pressure to give 420 mg, 63.5% of **1d** as a white solid. ^1^H NMR (400 MHz, CDCl_3_): δ 7.70–7.75 (m, 4H), 7.24–7.27 (m, 4H), 6.83 (s, 1H), 4.91 (s, 2H), 2.36 (s, 3H), 1.27 (s, 12H). ^13^C NMR (101 MHz, CDCl_3_): δ144.91, 138.23, 134.94, 133.62, 129.76, 128.59, 128.41, 83.92, 79.23, 24.88, 21.68. ESI-MS (positive mode) (*m*/*z*) calculated mass for C_20_H_26_BNO_5_SNa [M+Na]^+^ 426.15, found mass 426.29.

***N*-(benzyloxy)methanesulfonamide (2a)**. *O*-benzylhydroxylamine hydrochloride (510 mg, 2.50 mmol) was added to a solution of methanesulfonyl chloride (210 µL, 2.71 mmol) in pyridine (12 mL). The reaction mixture reacted overnight at room temperature and was quenched with 2M HCl (60 mL) and extracted with ethyl acetate (60 mL). The organic layer was washed with water (40 mL), brine (40 mL), dried over MgSO_4_ and concentrated using rotary evaporator to give 404 mg, 80.4% yield of **2a** as a white solid. ^1^H NMR (400 MHz, CDCl_3_): δ 7.37–7.41 (m, 5H), 6.86 (s, 1H), 5.00 (s, 2H), 3.04 (s, 3H). ^13^C NMR (101 MHz, CDCl_3_): δ 135.02, 129.52, 128.96, 128.88, 79.66, 37.03. ESI-MS (positive mode) (*m*/*z*) calculated mass for C_8_H_11_NO_3_SNa [M+Na]^+^ 224.06, found mass 224.01.

***N*-((4-nitrobenzyl)oxy)methanesulfonamide (2b)**. 1-[(Aminooxy)methyl]-4-nitrobenzene hydrochloride (613 mg, 3.0 mmol) was added to a solution of methanesulfonyl chloride (210 µL, 2.71 mmol) in pyridine (12 mL). The reaction mixture reacted overnight under room temperature and was quenched with 2M HCl (50 mL) and extracted with ethyl acetate (50 mL). The organic layer was washed with water (30 mL), brine (30 mL), dried over MgSO_4_ and concentrated using reduced pressure to give 516 mg, 77.5% yield of **2b** as white solid. mp 111–113 ⁰C; IR (film) 3211, 1608, 1517, 1320, 1151 cm^−1^; ^1^H NMR (400 MHz, CDCl_3_): δ 8.17 (d, *J* = 8 Hz, 2H), 7.49 (d, *J* = 8 Hz, 2H), 7.10 (s, 1H), 5.03 (s, 2H), 3.03 (s, 3H). ^13^C NMR (101 MHz, CDCl_3_): δ 148.13, 142.19, 129.74, 123.84, 78.08, 37.09. ESI-MS (positive mode) (*m*/*z*) calculated mass for C_18_H_10_N_2_O_5_SNa [M+Na]^+^ 269.23, found mass 269.06.

***N*-((4-azidobenzyl)oxy)methanesulfonamide (2c)**. *O*-(4-azidobenzyl) hydroxylamine (1.3g, 7.93 mmol) was added to a solution of methanesulfonyl chloride (675 µL, 8.72 mmol) in pyridine (30 mL). The reaction mixture reacted overnight at room temperature and was quenched with 2M HCl (150 mL) and extracted with ethyl acetate (150 mL). The organic layer was washed with water (100 mL), brine (100 mL), dried over MgSO_4_ and concentrated using rotary evaporator to give a crude product that was purified using column chromatography (Pet Ether/Ethyl Acetate, 3:1) to give 530 mg, 27.6% yield of **2c** as a white solid. mp 108–110 ⁰C; IR (film) 3207, 3024, 2943, 2111, 1505, 1304, 1157 cm^−1^; ^1^H NMR (400 MHz, CDCl_3_): δ 7.38 (d, *J* = 10 Hz, 2H), 7.04 (d, *J* = 10 Hz, 2H), 6.98 (s, 1H), 4.96 (s, 2H), 3.05 (s, 3H). ^13^C NMR (101 MHz, CDCl_3_): δ 140.78, 131.71, 131.17, 119.19, 78.91, 37.03. ESI-MS (positive mode) (*m*/*z*) calculated mass for C_8_H_11_NO_3_SNa [M+Na]^+^ 265.04, found mass 265.03.

**1-azido-4-methylbenzene (3)**. Was prepared following a published procedure [23]. A solution of sodium nitrite (6.3g, 92.8 mmol) in 26 mL of cold water and a solution of sodium azide [25] (6.1g, 92.8 mmol) in 58mL of cold water were added slowly to a solution of *p*-toluidine (10g, 92.8 mmol) in 50mL HCl: H_2_O (1:1) at 0 °C. After 1 h of reaction, the mixture was extracted with CHCl_3_ (100 mL), the remaining aqueous layer was washed with CHCl_3_ (2 × 50 mL) and the combined organic layers were washed with water (100 mL), brine (100 mL), dried over MgSO_4_ and concentrated under reduced pressure to give 9.5 g, 79.3% yield of **3** as a brown oil. ^1^H NMR (400 MHz, CDCl_3_): δ 7.16 (d, *J* = 8 Hz, 2H), 6.93 (d, *J* = 8 Hz, 2H), 2.33 (s, 3H). ^13^C NMR (101 MHz, CDCl_3_): δ 137.15, 134.63, 130.35, 118.85, 20.85.

**1-azido-4-(bromomethyl)benzene (4)**. Was prepared following a published procedure [25] *N*-Bromosuccinimide (10.51 g, 59 mmol) and 2,2′-Azobis(2-methylpropionitrile) (1 g, 6.1 mmol) were added under N_2_ atmosphere to a solution of 1-azido-4-methylbenzene (8 g, 60.1 mmol) in benzene (200 mL). The reaction mixture was heated under reflux for 3 days and concentrated under reduced pressure. The resulting product was partitioned between CH_2_Cl_2_ (100 mL) and washed with water (100 mL), brine (2 × 50 mL), dried over MgSO_4_ and concentrated to give a crude product that was purified by column chromatography (petroleum ether) to give 3.1g, 24.6% yield of **4** as a yellow-orange oil. ^1^H NMR (400 MHz, CDCl_3_): δ 7.40 (d, *J* = 12 Hz, 2H), 7.02 (d, *J* = 12 Hz, 2H), 4.50 (s, 2H). 

**2-((4-azidobenzyl)oxy)-isoindoline-1,3-dione (5)**. Was prepared following a published procedure [25] K_2_CO_3_ (2.34 g, 16.9 mmol) was added slowly to a solution of 1-azido-4-(bromomethyl)benzene (3.1 g, 14.6 mmol) and *N*-Hydroxyphthalimide (2.76 g, 16.9 mmol) in 45 mL DMF. The reaction mixture reacted overnight at room temperature and partitioned between CH_2_Cl_2_ (150 mL) and H_2_O (100 mL). The organic layer was washed with brine (100 mL), dried over MgSO_4_ and concentrated under reduced pressure to give 4.2g, 84.5% yield of **5** as a white solid. ^1^H NMR (400 MHz, CDCl_3_): δ 7.75–7.76 (m, 4H), 7.45 (d, *J* = 8 Hz, 2H), 6.96 (d, *J* = 8 Hz, 2H), 5.11 (s, 2H). ^13^C NMR (101 MHz, CDCl_3_): δ 163.51, 141.17, 134.51, 131.57, 130.41, 128.83, 123.57, 119.14 79.11.

***O*-(4-azidobenzyl) hydroxylamine (6)**. Was prepared following a modified procedure [25] Hydrazine monohydrate (1.04 mL, 32.4 mmol) was added slowly to a solution of 2-((4-azidobenzyl)oxy)-isoindoline-1,3-dione (4.2 g, 14.3 mmol) in CH_2_Cl_2_ (45 mL) and the reaction mixture reacted overnight at room temperature. The resulting mixture was quenched with 1M NaOH (50 mL) and the separated organic layer was washed with water (30 mL), brine (30 mL), dried over MgSO_4_ and concentrated under reduced pressure to give 1.78g, 76.4% yield of **6** as a yellow oil. ^1^H NMR (400 MHz, CDCl_3_): δ 7.27 (d, *J* = 8 Hz, 2H), 6.94 (d, *J* = 8 Hz, 2H), 5.35 (broad s, 2H), 4.56 (s, 2H). ^13^C NMR (101 MHz, CDCl_3_): δ 139.75, 134.29, 130.01, 119.06, 77.24. 

**2-(4-(4,4,5,5-tetramethyl-1,3,2-dioxaborolan-2-yl)phenoxy)isoindoline-1,3-dione (7)**. Was prepared following a published procedure [22]. Diethyl azodicarboxylate (1 mL, 5.2 mmol) was added slowly at 0 °C to a solution of triphenylphosphine (1.23 g, 4.7 mmol), *N*-hydroxyphthalimide (0. 77g, 4.7mmol), and 4-hydroxymethylphenylboronic acid pinacol ester (1 g, 4.3 mmol) in CH_2_Cl_2_ (10 mL). After 1 h of reaction, the reaction mixture was concentrated under reduced pressure and purified by column chromatography (Pet Ether/Ethyl Acetate, 10:1) to give 900 mg, 50.5% yield of **7** as a white solid. ^1^H NMR (400 MHz, CDCl_3_): δ 7.75–7.71 (m, 4H), 7.66–7.62 (m, 2H), 7.45 (d, *J* = 8 Hz, 2H), 5.15 (s, 2H), 1.26 (s, 12H). 

***O*-(4-(4,4,5,5-tetramethyl-1,3,2-dioxaborolan-2-yl)benzyl)hydroxylamine (8)**. Was prepared following a modified procedure [22]. Hydrazine hydrate (120 µL, 3.75 mmol) was added slowly to a solution of 2-(4-(4,4,5,5-tetramethyl-1,3,2-dioxaborolan-2-yl)phenoxy)isoindoline-1,3-dione (900 mg, 3.61 mmol) in ethanol (20 mL). The reaction mixture reacted overnight and the solid was filtered and rinsed with cold ethanol. The filtrate was concentrated under reduced pressure to give 410 mg, 45.6% yield of **8** as a yellow oil. ^1^H NMR (400 MHz, CDCl_3_): δ 7.64 (d, *J* = 8 Hz, 2H), 7.45 (d, *J* = 8 Hz, 2H), 5.15 (s, 2H), 1.26 (s, 12H).

### 3.2. Gas Chromatographic N_2_O Analysis

For headspace analysis, substrate (**1a**–**d**, **2a**–**c**, 0.04 mmol) was placed in a 10 mL round-bottom flask with a stir bar and sealed with a rubber septum. Solvent (methanol:water or methanol:PBS (100 mM), pH = 7.4; 1:1, 2 mL) was added using a syringe, and headspace aliquots (0.1 mL) from each experiment were injected at 1 and 24 h onto a 7890A Agilent Technologies gas chromatograph equipped with a micro-electron capture detector and a 30 m × 0.32 m (25 μm) HP-MOLSIV capillary column. The oven was operated at 200 °C for the duration of the run (4.5 min). The inlet was held at 250 °C and run in split mode (split ratio 1:1) with a total flow (N_2_ as carrier gas) of 4 mL/min and a pressure of 37.9 psi. The μECD was held at 325 °C with a makeup flow (N_2_) of 5 mL/min. The retention time of nitrous oxide was 3.4 min, and yields were calculated based on a standard curve for nitrous oxide gas (Gasco Precision Calibration Mixtures). To some samples, sodium borohydride (1.1 equivalents), sodium borohydride (1.1 equivalents) and copper (II) sulfate (0.1 equivalents), GSH (5 equivalents) or sodium sulfide (2–5 equivalents) were added. For oxidation of **1d**, hydrogen peroxide (5 equivalents) was added to a solution (2 mL) of the substrate in a mixture of 3:2:0.5 acetonitrile, water, PBS (100 mM, pH = 7.4).

## 4. Conclusions

Nitroxyl (HNO) demonstrates a unique biological profile compared to NO that deserves more detailed studies that have been afforded to other nitrogen oxides and gasotransmitters. The high chemical reactivity of HNO requires the use of donors and relatively complex detection methods, which complicate such studies and the confirmation of endogenous HNO production. HNO exhibits rich redox reactivity with thiols, hydrogen sulfide and heme proteins, suggesting a potential role in various redox-mediated processes. We report a small library of derivatives (**1a**–**d** and **2a**–**c**) of the HNO donor, Piloty’s acid, that liberate HNO upon oxidation or reduction through a 1, 6 elimination mechanism. These compounds were quickly constructed by the condensation of a sulfonyl chloride and an appropriately substituted hydroxylamine derivative. The nitro and azide-containing molecules (**1b**, **c** and **2b**, **c**) demonstrated HNO release, as measured by headspace GC for N_2_O, upon chemical treatment with reducing agents. Similarly, a boronate ester derivative (**1d**) generates HNO upon hydrogen peroxide oxidation. Control experiments with **1a** and **2a** show that only compounds with redox active groups generate HNO under these conditions, supporting the mechanism. Overall, these results show the ability of these redox-sensitive HNO donors to release HNO upon oxidation/reduction, and could find use in further defining the role of HNO in redox-based biological processes.

## Data Availability

Data available in Appendix A.

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
