# Peer review of "Para-Substituted O-Benzyl Sulfohydroxamic Acid Derivatives as Redox-Triggered Nitroxyl (HNO) Sources"

_molecules, 2022, doi:10.3390/molecules27165305_

Round 1
Reviewer 1 Report
In my opinion, the work is routine and of little interest to a wide audience of the Journal readers. The introduction is poorly connected with the synthetic part of the paper. I would recommend to submit the paper to a specialized journal making the accent on the pharmacological or biological problems - chemically the paper is of low interest.
Author Response
We appreciate the time the reviewer spent in reading/commenting on our manuscript. While this reviewer did not support publication in Molecules, we have added sections and references in the introduction to better explain the need and importance of redox-controlled HNO donors. We would also emphasize this is the first report of HNO donors that depend on oxidation or reduction of properly constructed substrates.
Reviewer 2 Report
King and coworkers report five para-substituted O-benzyl sulfohydroxamic acid derivatives that act as masked donors of nitroxyl (HNO), an unstable and biologically important small molecule. The design of these HNO releasing compounds is based on the generation of Piloty’s acid analogs, known HNO sources, through 1,6-elimination of p-quinone methide upon either oxidation (of boronate ester derivative, 1d) or reduction (of nitro and azide derivatives, 1b-c, 2b-c). The generation of HNO has been confirmed and estimated by gas chromatographic analysis of the reaction headspace for nitrous oxide content (the product of HNO dimerization and decomposition).
The redox-triggered nitroxyl donors presented in this work overcome some of the issues of conventional HNO donors such as Angeli’s salt (gives nitrite byproduct) and Piloty’s acid (decomposes at neutral pH). It is an interesting contribution to the field of HNO donors and these molecules are of potential interest in the biochemistry of nitroxyl. After addressing the comments listed below, I think this work is suitable for publication in Molecules.
Comments/suggestions:
1) It would be useful for potential readers of this paper to include in the introduction other examples of HNO donors used in biology and give a brief comparison discussing their pros and cons. For examples see: https://doi.org/10.1016/j.bbabio.2009.04.015
2) I wonder if there are any safety precautions that need to be taken into consideration in the synthesis and handling of the HNO donors reported herein (especially of azides 1c and 2c)? If any, a safety note should be added.
3) The time at which the headspace has been analyzed should be provided in Table 2 and in the experimental part. I could not find any discussion of the rate of decomposition. I suggest giving at least a qualitative description of how rapid or slow the reactions are.
4) Schemes 2 and 3 should be corrected. Some structures of O-benzyl hydroxylamines are incorrectly drawn with linear N-O-C angles!
5) In Scheme 3, correct the typo ‘benezene’ to ‘benzene’.
6) In Table 1, correct ‘2a-d’ to ‘2a-c’.
7) Page 5, line 157: in ‘Treatment of 1 or 2b’, specify which compound 1 has been used.
8) In the experimental section, the numbering of compounds 3–8 does not match with those in the main text!
9) Page 8, part 4.1.2: The volumes of the solvents and times of the reactions should be provided.
10) Page 6, Materials and Methods section states 1H NMR spectra were recorded using 300 MHz instrument, but NMR data below are all reported with 400 MHz…
Author Response
The authors would like to thank the reviewer for their comments supporting this paper and their very useful suggestions that have been incorporated. We feel their comments have greatly strengthened our manuscript.
Responses
1) We have added the suggested reference and modified the introduction to better describe the positive and negative characteristics of HNO donors.
2) Safety precautions for handling alkyl azides and sodium azide have been added to the Materials and Methods Section
3) A qualitative description of reaction rates has been added in Table 2 (results at 1 and 24 h) as suggested. The amounts of N2O formed increase over time and likely reflects the kinetics of the oxidation/reduction trigger reaction.
4 and 5) Schemes have been updated
6 and 7) Typos corrected
8) The numbering of the compounds from the manuscript to the materials and methods section has been fixed.
9) Volumes/times of reaction have been added to the Materials and Methods Section
10) Typo corrected
Reviewer 3 Report
see attachment

Author Response
We also would like to thank this reviewer for their time and suggestions that we feel strengthen this paper. Specific suggested changes:
1) Reference https://doi.org/10.1089/ars.2011.3937 has been added to the introduction. The other reference suggested was not add as this review dealt with metal heme complexes of HNO as traps (not donors).
2) A section of the reactivity of these compounds with E. coli nitroreductase (page 5, line 168) has been added. While this seems likely, our work and previous studies show HNO reacts with NADPH/NADH giving hydroxylamine and oxidized co-factor. While it appears compounds 1b and 2b can act as nitroreductase substrates, the product HNO is not stable under these reaction conditions.
3) The compound numbers from the manuscript to the experimental (materials and methods section) have been corrected and made consistent.
Reviewer 4 Report
Please, see the attached file.

Author Response
The authors want to thank this reviewer for their careful consideration and review of our paper and their very useful suggestions that have been incorporated. We feel their comments have greatly strengthened our manuscript.
Responses
1) This typo has been corrected
2) We believe this is the first example of using a 1, 6 elimination for the generation of HNO and this has been noted along with a supporting reference.
3) Phenyl sulfinic acid is the proposed by-product and has been added.
4) Reaction temperatures and yields (as well as style) have been changed in Scheme 3.
5) A variety of conditions have been reported for the condensation of sulfonyl chlorides and O-substituted hydroxylamines as the reviewer notes. The method we used is most similar to that in reference 33 (patent) and we are not completely sure this method is new.
6) Corrected
7) Added to Table 2
8) Following this reviewer's suggestion, we did analyze the reduction of 1b with sodium borohydride and identified p-amino benzyl alcohol by TLC and MS supporting the 1, 6 elimination pathway. These results have been added to the text.
9) We agree with the reviewer that "relative symmetry" and "poor solubility" are vague terms. The solubility of this compound has been better defined and it was analyzed under conditions that it was completely dissolved and this has been noted in the text (along with the removal of the vague terms).
10) We believe this language has been clarified as well. An advantage of Piloty's acid as an HNO source is that other groups can be appended to the aromatic ring and the base molecule retains the ability to generate HNO. The addition of nitro, azide and boronate groups to the O-benzylhydroxylamine group should have little effect on the HNO releasing ability of Piloty's acid or pCH3-C6H4SO2HNON used in these studies.
11) Numbering of compounds in the manuscript, Materials and Methods and SI have been standardized.
12) Both mp and IR spectra have been added for 2b and 2c
13) We greatly appreciate the reviewer's work here and acknowledge our presentation error. Reporting of the NMR data in the Materials and Methods section have been modified to correctly identify multiplicities along with coupling constants. Spectra have been re-analyzed with peaks correctly identified and not all as "multiplets". The spectrum of 5b, while contaminated with residual ethyl acetate, matches well with that published..
Round 2
Reviewer 1 Report
The content of the paper is rather far from my current research interests, so, I can anly repeat that, in my opinion, the paper is of low interest for chemists and should be transferred to a more proper specialized journal.
Reviewer 4 Report
Concerning schemes 2 and 3, I noticed that the authors made the suggested changes only to scheme 3 and not scheme 2 (See comments 4 and 6). I believe it has gone unnoticed, as it is a simple but important change.